# Juvenile Primary Fibromyalgia Syndrome: Advances in Etiopathogenesis, Clinical Assessment and Treatment: A Narrative Review

**DOI:** 10.3390/biomedicines13051168

**Published:** 2025-05-10

**Authors:** Claudio Lavarello, Silvana Ancona, Clara Malattia

**Affiliations:** 1Department of Neuroscience, Rehabilitation, Ophthalmology, Genetics, Maternal and Child Health (DINOGMI), Università degli Studi di Genova, 16132 Genova, Italy; lavarellocla@gmail.com; 2UOC Reumatologia e Malattie Autoinfiammatorie, IRCCS Istituto Giannina Gaslini, 16147 Genova, Italy; silvana.ancona@gmail.com

**Keywords:** juvenile fibromyalgia, juvenile primary fibromyalgia syndrome, chronic pain, nociplastic pain, central sensitization, sleep disturbances, cognitive-behavioral therapy

## Abstract

Juvenile Primary Fibromyalgia Syndrome (JPFS) is a complex, multifactorial condition characterized by widespread musculoskeletal pain, often accompanied by sleep disturbances, headaches, cognitive and mood disorders, and fatigue, resulting in a significant impact on the quality of life for affected children, adolescents, and their families. Although recent advances have improved the understanding of the underlying pathophysiological mechanisms and therapeutic approaches, its etiology and optimal treatments remain largely unknown. In this review, we summarize recent advances in the etiopathogenesis, clinical assessment, and treatment of JPFS. Our aim is to support clinicians in the diagnosis and management of JPFS patients, while also highlighting key areas that require further research to improve diagnostic accuracy and therapeutic outcomes.

## 1. Introduction

As initially described by Yunus and Masi in 1985, Juvenile Primary Fibromyalgia Syndrome (JPFS) is a disabling condition characterized by widespread chronic musculoskeletal pain that persists for longer than 3 months [1]. It is commonly accompanied by sleep disturbances, fatigue, cognitive impairment, anxiety, depression, and other somatic symptoms, with onset occurring before the age of 18 [1]. JPFS is more common in girls, with an estimated prevalence ranging from 0.5% to 6.2% [2,3,4,5,6,7]. The mean age at onset is between 11.4 and 13.7 years, although it is likely underdiagnosed in younger children [8].

Regarding sex differences, current literature suggests that males and females differ in their pain responses. Greater pain sensitivity and an increased risk of clinical pain are more commonly observed in females. However, more severe pain experiences, heightened pain-related distress, and a tendency towards catastrophic thinking about pain have been reported in males [9]. Additionally, a higher prevalence of depression and anxiety has been demonstrated among males [10].

JPFS often persists into adulthood [11], underscoring the importance of early recognition and intervention. Notably, it is associated with significant morbidity, including poor school attendance and difficulties in physical and psychosocial functioning [12], placing a considerable economic burden on healthcare systems [13]. In recent years, significant advancements have been made in understanding its etiopathogenesis and treatment. However, its underlying mechanisms remain incompletely understood, and further research is needed to explore potential treatment strategies. The absence of guidelines specifically tailored to children and young people further complicates JPFS management, necessitating a multidisciplinary therapeutic approach.

Diagnosis is also particularly challenging, as no specific biomarkers are currently available, and physical examinations often fail to reveal clear abnormalities. Consequently, patients are frequently referred to multiple specialists prior to receiving a diagnosis, resulting in unnecessary investigations and increased healthcare costs.

Given the diagnostic and therapeutic challenges commonly encountered in pediatric patients with JPFS, this narrative review aims to summarize recent findings on the etiopathogenesis, clinical assessment, and treatment of JPFS. By providing an updated overview of current evidence, it intends to support clinicians in delivering effective, evidence-based care and to highlight areas where further research is needed.

## 2. Methods

A narrative literature review was conducted, focusing on publications from the past 20 years. Searches were carried out in the PubMed database in February 2025 using the following search terms: “juvenile primary fibromyalgia syndrome”, as well as combined searches including “fibromyalgia” AND “children”, “fibromyalgia” AND “child”, and “fibromyalgia” AND “Pediatrics”. In addition, the reference lists of the included articles were examined to identify further relevant sources.

Only peer-reviewed articles published in English between January 2005 and February 2025 were considered. Exclusion criteria comprised non-English publications, articles not available in full text, and publication types such as case reports, editorials, letters to the editor, and conference abstracts.

## 3. Etiopathogenesis

JPFS is a multifactorial condition likely resulting from genetic predisposition combined with external stressors, such as emotional or physical trauma. These factors contribute to altered pain processing in the central nervous system (CNS), a phenomenon known as central sensitization, which may also involve peripheral mechanisms [14]. Although recent advances have improved the understanding of the underlying pathophysiological mechanisms, its etiology remains largely unknown [15,16].

### 3.1. Alteration in Pain Processing

Among the three categories of pain—nociceptive, neuropathic, and nociplastic—JPFS is primarily associated with nociplastic pain. Unlike nociceptive pain, which acts as an alarm system in response to a potentially harmful stimulus, and neuropathic pain, which results from nerve damage, nociplastic pain arises from alterations in nociceptive processing despite no detectable nerve or tissue injury [17]. These changes lead to increased sensitivity in the neural mechanisms responsible for distinguishing painful from non-painful stimuli, a characteristic feature of JPFS.

#### 3.1.1. Central Sensitization

Central sensitization is a complex phenomenon resulting from altered synaptic transmission within the CNS, involving receptors, neurotransmitters, ion channels, and signaling pathways. As a result, non-nociceptive stimuli are perceived as painful (allodynia), while noxious stimuli provoke an exaggerated, prolonged, and widespread pain response (hyperalgesia) [18].

Beyond the previously documented increased sensitivity to noxious pressure in adolescents with JPFS [19], a recent study reported heightened pain-evoked sensorimotor cortical responses in 33 adolescent girls with JPFS compared to healthy peers. Participants underwent functional magnetic resonance imaging (fMRI) scans before and after the application of noxious pressure to the left thumbnail. Affected individuals reported significantly higher pain intensity at both low and moderate pressure levels and exhibited increased activation in the right primary somatosensory cortex (S1) compared to the control group. Notably, peak S1 activation magnitude directly correlated with Widespread Pain Index (WPI) scores [20]. Similar findings in adult fibromyalgia (FM) patients [21] suggest that S1 plays a key role in pressure-evoked hyperalgesia in both juvenile and adult FM. However, differences in brain development between adults and younger individuals mean that findings from adult studies cannot be directly extrapolated to JPFS.

Resting-state functional connectivity (rsFC) is a neuroimaging technique that measures synchronized brain activity at rest and has recently provided insights into JPFS pathogenesis. A study by Suñol et al. (2024) examined rsFC and pain distribution in 37 female adolescents with JPFS compared to healthy peers [22]. The study found reduced rsFC within the paracentral lobule (PCL)/S1, a key sensorimotor cluster, and the left midcingulate cortex (MCC) [22], which is involved in acute pain processing and cognitive responses to pain [23]. Lower rsFC in these regions matched reported pain locations and predicted symptom severity. In contrast, the PCL/S1 showed increased connectivity with the thalamus, which processes sensory signals, and the right anterior insula, linked to emotion and salience. This may indicate heightened sensory input to the somatosensory cortex alongside reduced integration within PCL/S1 or impaired cortico-thalamic top-down inhibition. These findings were independent of symptom duration, suggesting that disrupted cortical connectivity at rest could be an early biomarker of JPFS [22].

A structural study comparing 34 female adolescents with JPFS to 38 healthy controls found reduced grey matter volume (GMV) in the anterior MCC [24]. Functional disability, fatigue, and pain interference correlated with increased GMV in inferior frontal regions, which are associated with affective, self-referential, and language-related processes. These findings partially overlapped with those observed in adult FM [25,26] and align with Suñol et al.’s functional study [22], reinforcing the idea that structural and functional alterations in the MCC may be a hallmark of both juvenile and adult FM.

Another resting-state fMRI study examined the role of resilience in JFPS pathogenesis and clinical presentation in 41 adolescent girls with JPFS [27]. Since JPFS-related brain regions overlap with those linked to resilience, particularly within the default mode network (DMN), the study explored whether resilience influenced symptom severity. The DMN plays a crucial role in self-related processing and attention shifting. The study found that the high-resilience JPFS group had fewer affective symptoms (e.g., depression and anxiety) but similar core somatic symptoms compared to the low-resilience group. Notably, they also showed stronger whole-brain and within-DMN connectivity, particularly in the anterior/dorsal posterior cingulate cortex (PCC), suggesting better regulatory function. In contrast, the low-resilience group had higher DMN-promotor connectivity. However, both JPFS groups displayed reduced PCL connectivity compared to healthy peers [27], a feature consistent with previous findings [22].

In conclusion, JPFS is characterized by alterations in the sensorimotor network, extending to other sensory systems, with notable involvement of S1, PCL, and MCC. These changes result in reduced cortico-cortical sensory integration and increased sensory-affective connectivity at rest, alongside structural changes in regions involved in emotional and language processing. Further longitudinal studies are needed to determine whether these structural and functional changes precede JPFS onset or emerge as pain-driven alterations. If confirmed, targeting cortical sensorimotor systems could be a potential therapeutic strategy in JPFS, as explored in adult FM patients [28].

#### 3.1.2. Small Fiber Pathology

While CNS dysfunction plays a primary role in JPFS, peripheral nociceptive input may also contribute to its pathogenesis. A meta-analysis of eight studies reported that 49% (95% CI 38–60%) of adults with FM exhibit small fiber pathology [29]. A subsequent study of 117 skin-biopsied FM patients found that greater symptom severity correlated with increased cutaneous denervation [30].

Similarly, Boneparth et al. (2021) examined lower-leg skin biopsies in 15 JPFS patients and compared their nerve fiber density to 23 matched healthy controls [31]. They found that 53% (8/15; 95% CI 26–79%) of JPFS patients had an epidermal neurite density (END) below the 5th centile of the age/gender/race-based normal range, compared to just 4% (1/23; 95% CI 0–22%) of healthy controls (*p* < 0.01). The mean END in JPFS patients was 273/mm^2^ (95% CI 198–389), significantly lower than the 413/mm^2^ (95% CI 359–467) observed in controls. However, no significant differences were found in pain scores, symptom duration, functional disability, or autonomic symptoms between JPFS patients with END above or below the 5th centile [31].

These findings resemble small fiber neuropathy (SFN), a condition affecting small A-delta and C fibers. However, the role of cutaneous denervation in JPFS remains unclear. Some researchers suggest that undiagnosed SFN could directly cause FM [30], while others argue that small fiber changes are secondary to centralized chronic pain mechanisms [32]. In contrast, Oaklander et al. propose that FM in such cases was simply misdiagnosed [33]. Given the different treatment approaches for FM and SFN, distinguishing between them is crucial in clinical practice. A recent review by Ahmed et al. provides guidance on this differentiation [34].

### 3.2. Autonomic Nervous System Dysregulation

A recent multicenter study by Lynch-Jordan et al. found that over 80% of adolescents with JPFS reported neurologic or autonomic symptoms [35], suggesting that autonomic nervous system dysregulation may contribute to its pathogenesis.

Previous research in a cohort of 25 adolescents with JPFS identified signs of autonomic dysfunction, including chronotropic incompetence (an inability to appropriately increase heart rate during exercise) and delayed heart rate recovery following exertion [36]. This dysregulation may contribute to symptoms such as dizziness, syncope, and exercise intolerance, further worsening physical impairment beyond the direct effects of chronic pain [35].

### 3.3. Sleep Disturbances

Sleep disturbances are highly prevalent in FM, affecting 75–90% of adults and 67–96% of JPFS patients [1,8,37,38,39]. This may be due to a shared pathophysiology, as neurotransmitters involved in pain processing also play a role in sleep regulation [40].

Objective sleep studies in JPFS—such as actigraphy, polysomnography, and sleep latency tests—have identified several disruptions. These include increased sleep-onset latency, more frequent awakenings, prolonged wake after sleep onset, reduced total sleep time and efficiency, more periodic limb movements, and increased alpha-delta intrusions into slow-wave sleep. Collectively, these findings suggest an underlying homeostatic and circadian dysregulation in JPFS [41,42,43].

A recent study by Malattia et al. compared 25 JPFS patients with 27 age-matched controls, using self-report measures and full-night polysomnography. The study found that 88% of JPFS patients reported non-restorative sleep, a finding that correlated with WPI scores. While total sleep time was similar between the two groups, JPFS patients experienced longer sleep periods and increased wake after sleep onset. The study also highlighted an altered distribution of N3 sleep, with more N3 epochs occurring in the second half of the night than expected. This suggests impaired homeostatic sleep regulation, which may contribute to morning symptoms. Despite discrepancies between subjective and objective sleep data, greater sleep disturbances were associated with more severe pain and depressive symptoms [39]. Interestingly, major sleep alterations seen in adult FM, such as reduced total sleep time, seem less pronounced in JPFS, as previously noted by Roizenblatt et al. [41]. This suggests that sleep disruption in JPFS may worsen over time. Malattia et al. also found that both objective and subjective sleep disturbances were correlated with depressive symptoms, proposing that serotoninergic dysregulation and altered melatonin secretion may contribute to both conditions [39]. However, further studies are required to clarify the bidirectional relationship between sleep disturbances and mood disorders in JPFS.

### 3.4. Genetic and Epigenetics

Genetic predisposition plays a key role in JPFS, as evidenced by familial aggregation. For instance, a Finnish twin cohort study on FM estimated its heritability at approximately 50% [44], which aligns with the well-documented familial clustering of the condition [45,46]. Similarly, Schanberg et al. found that 79% of the parents of JPFS children reported a history of at least one chronic pain condition [47].

Epigenetic factors may also play a role, as research has linked FM to hypomethylation of genes involved in DNA repair, autonomic regulation, and subcortical neuronal function [48]. However, gene-environment interactions in JPFS remain largely unexplored.

### 3.5. Psychological and Familiar Factors

Altered pain perception, influenced by psychological factors, can reduce an individual’s ability to cope with pain [49]. Anxiety and depression are common among young people with chronic pain and tend to be more pronounced in JPFS patients. These psychological symptoms are associated with increased disability, poorer functioning, higher school absenteeism, and more frequent healthcare visits [50]. However, a recent study of 37 adolescents (21 with chronic pain from headaches or recurrent non-inflammatory, non-neoplastic arthralgia, and 16 with JPFS) found no significant differences between the two groups in depression, anxiety, school absenteeism, medication use, or reported pain levels [51].

Children and adolescents with widespread pain often come from families where parents have a history of pain, FM, or mental health disorders. This familial influence can contribute to poor coping mechanisms, increasing the risk of anxiety and depression, which may, in turn, contribute to JPFS onset. Anxious or disorganized family environments may increase JPFS risk, while a controlling family dynamic in early adolescence is linked to poorer long-term emotional well-being and a higher likelihood of developing depressive symptoms [50,52,53]. A recent Italian study supported these findings, showing that parents of JPFS patients were more likely to have lived apart, experienced diffuse pain, been diagnosed with FM (odds ratio (OR) of 2.54 for children with at least one affected parent), or used psychotropic medications [7].

A history of trauma is also common in JPFS patients and is linked to a higher risk of psychological comorbidities. However, trauma does not appear to correlate with more severe pain or reduced physical functioning [54]. Furthermore, higher resilience has been shown to be a protective factor against affective symptoms, though it did not affect core somatic symptoms [27].

### 3.6. Hypermobility

Joint hypermobility (JH) is commonly observed in JPFS patients, with prevalence estimates ranging from 40% to 81% [38,55]. Increased joint laxity is believed to alter compensatory biomechanisms, potentially contributing to physical impairments. Repeated microtrauma, occasional joint dislocation, and an elevated risk of injury may lead to persistent localized pain, particularly in FM patients with JH [56].

In a cohort of 131 JPFS patients, Ting et al. (2012) found that those with JH exhibited greater pain sensitivity, as evidenced by lower pressure pain thresholds and a higher number of painful tender points (TPs) compared to non-JH patients. However, no significant differences were observed in self-rated pain intensity [57].

A more recent study by Black et al. (2023) [58] assessed pain and biomechanics in JPFS adolescents, comparing those with and without JH during a moderately vigorous landing and jumping task. Among 36 JPFS patients (13 with JH, 23 without), no differences in pain levels were found, although the JH group demonstrated lower overall physical impairment. Small but significant biomechanical differences were noted, particularly in hip positioning and movement [58], suggesting that tailored exercise programs could be beneficial.

Additionally, Malattia et al. identified a higher prevalence of temporomandibular joint (TMJ) hypermobility in JPFS patients compared to healthy controls. In a study of 30 JPFS patients and 45 controls, the JPFS group showed greater mouth opening width, indicating TMJ hypermobility. This was associated with increased pain sensitivity in the orofacial region. TMJ hypermobility (OR 1.42, CI 95% 1.10–1.84), orofacial pain (OR 21.0, CI 95% 2.56–173.0), and diffuse tenderness in the masticatory muscles (OR 14.9, CI 95% 1.38–160.8) were independently linked to JPFS [59].

## 4. Diagnosis

### 4.1. Clinical Manifestation

The hallmark symptom of JPFS is diffuse, chronic pain, often described as deep, continuous, and predominantly axial [35]. This pain tends to worsen with psychological stress, physical activity, and infections [60]. JPFS patients report higher levels of pain in terms of frequency, intensity, and interference compared to those with other rheumatic conditions, such as juvenile idiopathic arthritis, or other forms of non-specific chronic pain [61,62]. As a result, they experience greater malaise, a reduced quality of life, and, notably, heightened pain-related stigma from peers [62]. These findings suggest that young people with JPFS may be more concerned about their condition than their peers with other autoimmune diseases, potentially leading to greater avoidance of daily activities, higher school absenteeism, and fewer social opportunities. A key clinical consideration is the importance of evaluating peer relationships when assessing pain in JPFS patients, as these serve as significant social markers during adolescence [62].

Other common symptoms include fatigue, sleep disturbances, mood disorders, cognitive changes, and various somatic symptoms.

Fatigue is reported by over 85% of patients, with more than 40% rating it as a severe symptom [7,35]. Sleep disturbances affect more than 80% of JPFS patients and contribute to fatigue and non-restorative sleep. Common issues include difficulty falling asleep, reduced sleep efficiency, frequent awakenings, and excessive daytime sleepiness. Over 30% of patients also report regular afternoon naps [39].

Psychiatric comorbidities are prevalent, with around 70% of children and adolescents with JPFS diagnosed with a psychiatric disorder, most commonly anxiety. These patients also experience higher levels of functional disability, though no significant differences in pain severity are observed compared to those without psychiatric comorbidities [63]. While major depression is less common in JPFS than in adult patients, children and adolescents with JPFS report higher levels of depression than peers with other forms of generalized pain [64]. Studies show that 26–28% of JPFS patients experience suicidal ideation [65], and nearly a third severe depressive symptoms [35].

Cognitive impairment is another concern. A recent study by Gmuca et al. (2022) found significant rates of cognitive dysfunction in 31 JPFS patients, with 65% reporting subjective cognitive difficulties and 39% demonstrating objective cognitive deficits. The most affected domains were psychomotor speed, executive function, and attention [66]. In a multicenter cohort described by Lynch-Jordan (2023), over half of the patients (61%) reported moderate to severe issues with concentration or memory [35]. Given these concerning findings, close psychological and neuropsychological monitoring is crucial to assess attention and mood symptoms. Comprehensive mental health support should undoubtedly be integrated into the multidisciplinary approach to JPFS, as recommended by Lynch-Jordan [35].

Headaches are also common in JPFS, affecting up to 80% of patients, regardless of the age of onset [35]. A recent study by Gupta et al. (2024) found a 31% prevalence of tinnitus in children with JPFS, a rate similar to prior studies in adults, though the exact causation remains unclear [67].

Other prevalent symptoms in JPFS include joint hypermobility, joint stiffness/swelling, and abdominal pain, with higher frequencies observed in children under ten years of age [68]. Adolescents, on the other hand, are more likely to report chronic or recurrent chest pain of non-cardiac origin [69].

Interestingly, recent studies have shown that JPFS patients often experience orofacial pain, including tenderness upon palpation, perceived jaw functional limitation, and pain-related disability, such as difficulties with eating and speaking. Given the high prevalence of temporomandibular disorders (TMDs) in JPFS (87%, according to Malattia et al.), a thorough evaluation of the stomatognathic system is now considered essential for both diagnosis and follow-up care [59].

### 4.2. Diagnosis

The clinical complexity of JPFS, along with its overlap with other conditions such as chronic fatigue syndrome, migraine, and irritable bowel syndrome (IBS), often makes diagnosis challenging. This can result in a diagnostic delay ranging from six months to five years after symptom onset [8]. Such delays can complicate the timely initiation of appropriate treatment, contributing to the chronic nature of symptoms, increased disability, and higher costs for patients, their families, and society.

Currently, the diagnosis of JPFS is based on clinical evaluation, requiring a comprehensive medical history. This should include an assessment of pain characteristics and location, as well as the presence and severity of associated symptoms, such as sleep and mood disorders. A thorough physical examination is essential to rule out arthritis or other musculoskeletal conditions. The examination should also assess central sensitization to pain, which can be evaluated through the bilateral acupressure response at the 18 TPs. However, this assessment is no longer mandatory for diagnosing JPFS due to challenges such as inconsistencies in technique, poor reproducibility between examiners, variations in applied pressure, individual differences in pain sensitivity, and generally lower acupressure sensitivity in male patients [70].

In 1985, Yunus and Masi proposed clinical criteria for JPFS (Table 1) [1], though these have not been validated.

Currently, the American College of Rheumatology (ACR) diagnostic criteria, initially validated in 2010 for adult FM [72] and revised in 2016 [73], are applied in both pediatric clinical practice and research. When applied to JPFS adolescents, the 2010 ACR FM criteria demonstrated a sensitivity of 89.4% and specificity of 87.5% [74].

The 2010 and 2016 ACR FM criteria include a clinician-administered tool for the WPI and a Symptom Severity (SS) scale to assess key characteristic symptoms (Figure 1).

The WPI comprises a list of 19 painful areas, representing the bodily distribution of pain (Table 2).

The SS scale is divided into two components (Figure 2). The first (SSa) measures the severity of fatigue, unrefreshing sleep, and cognitive symptoms using a 4-point Likert scale, with a total score ranging from 0 to 9. The second (SSb) consists of a checklist of 41 somatic symptoms, scored from 0 to 3 based on the number of symptoms reported by the patient: 0 symptoms (score of 0), 1 to 10 symptoms (score of 1), 11 to 24 symptoms (score of 2), and 25 or more symptoms (score of 3). The final SS score is calculated by summing the SSa and SSb scores, resulting in a total score between 0 and 12.

### 4.3. Clinical Assessments

Developing an optimal therapeutic strategy for JPFS requires sensitive and accurate tools to assess its severity. However, there is a notable lack of instruments specifically designed for this condition. Most assessment measures used for adults with FM are unsuitable for pediatric patients, as they include activities typically not performed by children, such as shopping, doing laundry, and driving [75].

To our knowledge, only one measure has been adapted for JPFS: the Modified Fibromyalgia Impact Questionnaire (FIQ)—Child version, which replaces references to “work” with “school” to enhance relevance for pediatric patients [76]. However, its psychometric properties and clinical utility have only been examined in a limited number of studies [76,77].

Several child-centered outcome measures are available for JPFS assessment, including the PROMIS Pediatric Pain Interference (PPI) Anxiety and Depression, the Functional Disability Inventory (FDI), and the PedsQL 3.0 Rheumatology Module [77,78]. However, none of these tools are disease-specific nor fully capture the condition’s overall burden. Given the multifaceted nature of JPFS, a comprehensive evaluation extending beyond pain and physical function is essential. As highlighted by the Outcome Measures in Rheumatology Clinical Trials Fibromyalgia Syndrome Workgroup, a multidimensional approach is necessary to fully understand the broad spectrum of symptoms and their impact on daily functioning [79].

A thorough clinical assessment should consider all potentially affected domains. Since sleep disturbances are a hallmark of JPFS, the Epworth Sleepiness Scale for Children and Adolescents (ESS-CHAD) [80] is a useful tool for assessing daytime sleepiness. However, due to the discrepancy between subjective and objective sleep measures [39], polysomnography remains a valuable tool for evaluating reported sleep disturbances, particularly when considering pharmacological treatment. Polysomnography provides an objective assessment of both quantitative and qualitative sleep alterations, enabling the identification of poor sleep efficiency, frequent awakenings, reduced total sleep time, and the presence of an alpha-delta sleep pattern [39,42,43].

As previously mentioned, close psychological and neuropsychological monitoring is crucial in the management of JPFS. Various tools can be used for psychological evaluation, such as the State Trait Anxiety Inventory (STAI-Y) [81] and the Multidimensional Anxiety Scale for Children (MASC) [82] for anxiety assessment. For depression assessment, the Children’s Depression Inventory (CDI) [83] and the Beck Depression Inventory (BDI-II) [84] are widely used.

Cognitive dysfunction should also be thoroughly assessed. Useful tools for intellectual ability evaluation include the Wechsler Intelligence Scale for Children—Fourth Edition (WISC-IV) for ages 6–16 [85], and the Wechsler Adult Intelligence Scale—Fourth Edition (WAIS-IV) for older patients [86]. The Behavior Rating Inventory of Executive Function—Second Edition (BRIEF-2) [87] is valuable for assessing executive function, including impulse control, cognitive flexibility, response modulation, and future planning. For the assessment of behavioral problems, the Youth Self Report (YSR) [88] for ages 11–18 is commonly used.

Motor development should be evaluated in JPFS patients, especially in those with concerns about motor delays. The Movement Assessment Battery for Children—Second Edition (MABC-2) is a useful tool for evaluating motor coordination and developmental delays [89].

Finally, as part of a multidisciplinary approach, a thorough gastrointestinal assessment should be conducted in patients presenting with gastrointestinal symptoms.

## 5. Treatment

The primary goal of treatment is to reduce pain and enhance quality of life. Given the complexity of JPFS, an individualized and multidisciplinary approach is essential. Equally important is early intervention to prevent pain and disability from becoming more entrenched and difficult to manage.

First and foremost, it is crucial to reassure both patients and their families, emphasizing that, although the pain is real and its duration cannot be precisely predicted, there are multiple treatment options available.

Due to the heterogeneous nature of symptoms, and in line with European League Against Rheumatism (EULAR) guidelines for adult FM [90], treatment should be tailored to the predominant symptoms affecting each patient [91]. However, current literature on JPFS treatment remains limited, focusing almost exclusively on non-pharmacological approaches, such as physical activity and cognitive-behavioral therapy (CBT), which should be considered first-line treatments.

Therapeutic management aims to alleviate the most debilitating symptoms through a combination of non-pharmacological and pharmacological treatments, often implemented in a stepwise approach. Patients respond best to a personalized and multidisciplinary treatment plan involving various healthcare specialists, including rheumatologists, neurologists, physiatrists, rehabilitation therapists, psychologists, and gastroenterologists.

Current evidence regarding JPFS treatment strategy is discussed below and summarized in Table 3.

### 5.1. Non-Pharmacological Treatment

The American Pain Society guidelines recommend moderate to vigorous aerobic exercise for at least 30 min, two to three times weekly, for children and adolescents with JPFS [90,91]. In a study by Stephens et al., a 12-week aerobic exercise program was compared with qigong. Both groups experienced symptom improvements, but the aerobic group showed greater benefits in fatigue, quality of life, and physical functioning [92].

Despite these positive outcomes, adherence remains a significant challenge. JPFS patients tend to be less physically active than their peers, as exercise is often associated with increased pain [100], leading to poor adherence to physical activity [101]. This reluctance has also been attributed to fear of movement, a common issue among JPFS patients. Such fear can lead to altered movements patterns, increasing the risk of further pain or injury and reinforcing the cycle of sedentary behavior [101].

Among non-pharmacological treatments for both FM and JPFS, the strongest evidence supports psychological therapies, particularly CBT. CBT is a goal-oriented psychotherapy that has been shown to improve pain, functional disability, and depression in the short term [93,102,103,104]. Additionally, it enhances coping efficacy and reduces catastrophizing in the long term, even after discontinuation [105]. However, despite significant self-reported improvements in functioning, CBT has not been associated with increased physical activity [100]. 

For this reason, recent research has focused on a combined approach that integrates CBT with physical activity, showing the most promising results as a treatment strategy for JPFS. One example is the Fibromyalgia Integrative Training program for Teens (FIT Teens), which combines CBT with a tailored neuromuscular exercise program aimed at improving psychological coping skills and movement competence while reducing fear of movement [101]. FIT Teens is part of an ongoing multi-site, three-arm randomized controlled trial (RCT). The intervention consists of an intensive 16-session program (twice weekly for eight weeks), delivered in small groups of 4–6 patients. Following a successful Phase 1, which demonstrated good tolerability [106,107], Phase 2 involved a pilot RCT comparing FIT Teens to CBT alone. This trial showed that FIT Teens was superior in reducing pain [94] and improving both strength and biomechanics [95]. Specifically, participants in the FIT Teens group exhibited better hip- and knee-related biomechanics without any treatment-related adverse events. These improvements resulted in greater stability during functional movements, increased movement competence, and a reduced risk of injury [95]. The ongoing Phase 3 RCT is a comparative effectiveness study designed to determine whether the combined FIT Teens intervention is superior to CBT alone or graded aerobic exercise alone [108]. Due to the COVID-19 pandemic, some adaptations to the protocol were necessary [109]. The results of this Phase 3 RCT will provide valuable insights into the effectiveness of non-pharmacological strategies for JPFS.

### 5.2. Pharmacological Treatment

Although various pharmacological strategies are commonly used for adult FM, no medications have yet been approved for the treatment of JPFS. Given the complete lack of specific evidence, we will briefly discuss the pharmacological options currently available for adult FM from a pediatric perspective.

A recently published overview of 21 Cochrane reviews evaluating pharmacological therapies for adult FM found that only pregabalin, duloxetine, and milnacipran demonstrated moderate-to-good evidence of substantial pain relief for a small proportion of patients (approximately 1 in 10) with moderate to severe FM pain over a period of 1–3 months. These drugs also showed a favorable safety profile, although there is no evidence supporting their efficacy beyond six months [110]. All three medications have been approved by the U.S. Food and Drug Administration for adult FM; however, they have not yet been licensed for this indication by the European Medicines Agency. No other drugs demonstrated significant efficacy for FM pain in this review. However, selective serotonin reuptake inhibitors (SSRIs) were found to be effective in treating depression, while mirtazapine showed benefits for sleep disturbances. Among serotonin-norepinephrine reuptake inhibitors (SNRIs), duloxetine and milnacipran were associated with improvements in fatigue, depression, anxiety, and overall quality of life [110].

#### 5.2.1. Gabapentinoids

Pregabalin and gabapentin are among the most frequently used drug for the treatment of adult FM, demonstrating both efficacy and good tolerability [111,112].

In 2016, a double-blind RCT evaluated the effects of pregabalin in 80 adolescents with JPFS over approximately one year (44 received pregabalin, 36 received placebo; mean age 14.7 ± 1.2 years). Surprisingly, the trend toward improvement in mean pain scores with pregabalin versus placebo was not statistically significant. However, significant improvements were observed in certain measures, including changes in pain scores and the patient global impression of change. Nevertheless, trends in other secondary outcomes, such as pain, sleep quality, and FM impact, did not reach statistical significance [96].

Regarding safety, the adverse event profile of pregabalin in JPFS was consistent with that observed in adult FM patients, with 70% of adolescents on pregabalin experiencing at least one adverse event, compared to 64% in the placebo group [96].

#### 5.2.2. Antidepressants

The pharmacological treatment of FM includes three classes of antidepressants: SNRIs, SSRIs, and tricyclic antidepressants (TCAs).

Evidence for the use of SNRIs in JPFS remains limited. A recent study involving 149 adolescents with JPFS, treated for approximately one year with duloxetine, found no significant improvement in average pain severity from baseline to week 13 compared to placebo. However, significantly more patients on duloxetine achieved a ≥30% and ≥50% reduction in pain severity compared to those on placebo [97]. It is worth noting that duloxetine was associated with a significantly higher incidence of treatment-emergent suicidal ideation, behavior, and other severe psychiatric adverse events compared to placebo [113].

An open-label study on milnacipran in JPFS showed improvements in pain and quality of life, with a good safety profile. However, the study was prematurely terminated due to low enrollment rates [98].

Regarding SSRIs, such as fluoxetine and paroxetine, evidence of their efficacy in adults with FM has been less convincing [114,115]. An open-label study of 10 female adolescents with JPFS, treated with fluoxetine, showed a reduction in pain and overall symptom improvement. However, patients only tolerated low doses of the drug, indicating a higher sensitivity to adverse effects compared to adults [99].

Amitriptyline, a TCA, is commonly used in FM based on evidence from adult populations. Although research on antidepressants for JPFS is limited, these medications play a crucial role in managing comorbid psychiatric conditions, such as anxiety and depression, which are frequently observed in JPFS patients [90,91].

#### 5.2.3. Cyclobenzaprine

Cyclobenzaprine, a muscle relaxant structurally similar to TCAs, is weakly recommended for the treatment of FM in adults with sleep disorders [90,91]. A recent study involving over 500 adult FM patients demonstrated that a sublingual, low-dose, slow-release formulation of cyclobenzaprine is safe, well tolerated, and effective, resulting in a significant reduction in pain intensity [116].

This therapeutic option may be particularly suitable for adolescent patients due to its reduced frequency of administration and sublingual formulation, which could improve adherence. However, specific studies on JPFS patients are needed to assess its safety and efficacy in this population.

#### 5.2.4. Cannabinoids

Cannabinoids may have a role in managing FM due to their potential effects on pain modulation and associated symptoms, particularly sleep disturbances, anxiety, and depression [117].

Currently, no clinical studies have evaluated the use of cannabis in pediatric patients with JPFS. However, a recent systematic review and meta-analysis assessed the effects of medical cannabinoids, primarily used for conditions such as epilepsy and chemotherapy-induced nausea and vomiting, in children and adolescents through RCTs. The findings indicated an increased risk of adverse events, particularly diarrhea, transaminitis, and somnolence [118]. Furthermore, cannabinoids seem to have a more hazardous effect on adolescents than on adults due to the plasticity and ongoing development of the brain during childhood [119]. Compared to adults, adolescents who begin using cannabis perform worse in cognitive tests, exhibiting deficits in memory, attention, and verbal fluency [120]. These safety concerns underscore the need for further research before considering cannabinoids as a potential therapeutic option for JPFS.

#### 5.2.5. Supplementary and Physical Therapies

The treatment of FM primarily relies on a non-pharmacological approach that combines physical exercise with psychological therapy, particularly CBT. In resistant cases, pharmacological treatments may be integrated. Additionally, several complementary therapies can enhance the effectiveness of standard treatments, including nutritional supplements, acupuncture, mind-body interventions, and nerve stimulation.

##### Nutritional Supplements and Adjunctive Therapies

Vitamin D supplementation has been considered an adjunctive therapy for JPFS [121,122]. Although the causal relationship between vitamin D deficiency and FM remains inconclusive, some studies suggest potential benefits. In a pilot study involving children with musculoskeletal conditions and chronic or recurrent pain associated with vitamin D deficiency, six months of vitamin D supplementation led to improvements in pain intensity and daily functioning [123]. However, large-scale randomized studies specifically focusing on JPFS patients are required to confirm these findings.

Palmitoylethanolamide (PEA) [124,125] and melatonin [126] have demonstrated effectiveness and safety in managing chronic pain conditions, including FM. PEA, a saturated fatty acid amide of palmitic acid, has been shown to reduce pain and positive TPs when used as an add-on treatment alongside duloxetine and pregabalin in adult FM patients [124]. A recent open-label study involving 50 FM patients treated with a daily combination of PEA and melatonin for four months reported reductions in pain intensity, improved sleep quality, and an overall enhancement in quality of life [127]. Regarding pediatric patients, PEA has only been proved effective in reducing abdominal pain in children with IBS [128]. Therefore, further research is needed to explore its potential benefit for JPFS.

##### Acupuncture and Mind-Body Interventions

Acupuncture has been shown to help manage various chronic pain conditions, including adult FM [129,130] and pediatric primary headaches [131,132]. However, further research is needed to confirm its benefits for JPFS.

Mindfulness-based meditation has been explored as a strategy to reduce pain catastrophizing in adolescents with chronic pain. A small pilot study suggested it could be beneficial, although results on other pain-related outcomes were inconsistent [133]. Recent evidence also indicates that mindfulness meditation can improve quality of life in FM patients by reducing stress, insomnia, and depression in the short and medium term. Additionally, mindfulness techniques have been effective in enhancing cognitive and emotional skills in children, making them a promising approach for JPFS.

Yoga and Tai Chi have been found to reduce FM symptoms in adults [134,135], including improvements in depression, anxiety, and sleep quality [136]. However, studies on their effects in JPFS are lacking, highlighting the need for further research.

Guided imagery and hypnosis have been effective in reducing pain and psychological distress in adult FM patients, but studies on their effectiveness in JPFS are still needed [137].

##### Physical Therapies

Physical therapies employ various techniques to alleviate pain and improve function in musculoskeletal and neuromuscular conditions.

Transcutaneous Electrical Nerve Stimulation (TENS) is a non-invasive method that uses low-voltage electrical currents to activate the body’s natural pain-relief mechanisms [91]. A recent meta-analysis found that TENS significantly reduced pain, disability, and improved quality of life in adult FM patients [138]. Interestingly, it has also recently been shown to be effective in controlling acute postoperative pain, significantly decreasing opioid (morphine) requirements, as well as the incidence of nausea and vomiting, dizziness, and pruritus [139].

Transcranial Direct Current Stimulation (tDCS) and Transcranial Magnetic Stimulation (TMS) are emerging techniques for treating chronic pain. In FM patients, tDCS applied to the motor cortex has provided short- and medium-term pain relief. Both tDCS and TMS have been shown to reduce pain sensitivity, catastrophizing, and improve quality of life, while stimulation of the prefrontal cortex has been associated with reduced fatigue. However, their effects on anxiety and depression remain unclear [140].

Furthermore, pulsed electromagnetic field therapy has demonstrated efficacy in the management of adult FM, particularly by reducing pain intensity, FM severity, and anxiety [141].

At present, no studies have been conducted on these techniques in JPFS patients, highlighting the need for further research.

### 5.3. Final Considerations on Treatment

Despite growing interest in the management of JPFS, the available literature remains limited, fragmented, and often of low methodological quality. Most existing studies involve small sample sizes, short follow-up periods, and, in some cases, lack adequate control groups. Furthermore, the variability in diagnostic criteria complicates direct comparison across findings. While the strongest evidence supports non-pharmacological strategies—particularly aerobic exercise, CBT, and their combination—major gaps remain concerning long-term efficacy and the sustainability of these interventions.

Low adherence to physical activity—often due to pain exacerbation and fear of movement—represents a significant clinical challenge. This highlights the need for personalized rehabilitation strategies and targeted psychological support. Although pharmacological therapies are commonly used in clinical settings, the evidence supporting their use in pediatric populations remains limited and sometimes inconsistent. Drug such as duloxetine and pregabalin have not yet demonstrated clinically significant benefits in pediatric RCTs and may be associated with concerning adverse effects.

The absence of pharmacological treatments specifically approved for JPFS, together with the lack of controlled studies on complementary or neuromodulatory interventions, underscores the urgent need for further research. Well-designed, multicenter RCTs with extended follow-up periods are essential to identify effective, sustainable, and safe approaches, as well as to explore predictive markers of treatment response. Ultimately, the primary goal remains to improve patients’ quality of life through a multidisciplinary, patient-centered approach grounded in robust scientific evidence.

## 6. Conclusions

JPFS is a complex, multifactorial condition marked by widespread chronic pain, often accompanied by symptoms such as fatigue, sleep disturbances, headaches, anxiety, depression, and gastrointestinal issues. Recent studies have identified functional and structural changes in the somatosensory cortex and MCC, though further longitudinal studies are needed to determine whether these alterations precede the onset of JPFS or result from pain-driven changes.

The clinical heterogenicity of JPFS, along with the lack of specific diagnostic criteria and treatment guidelines for children and adolescents, presents significant challenges in diagnosis and management.

This review aims to provide clinicians with tools to monitor and manage JPFS through a multidisciplinary approach. As recent research indicates, non-pharmacological strategies, particularly the combination of CBT with tailored neuromuscular exercise training, have proven more effective in reducing pain and improving strength and biomechanics compared to CBT alone.

While the available evidence is expanding, further research is essential to deepen our understanding of the pathophysiology of JPFS and refine treatment strategies. Specifically, there is a clear need for more focused studies in pediatric populations to establish evidence-based pharmacological and complementary treatments and to develop clearer diagnostic guidelines.

## Figures and Tables

**Figure 1 biomedicines-13-01168-f001:**
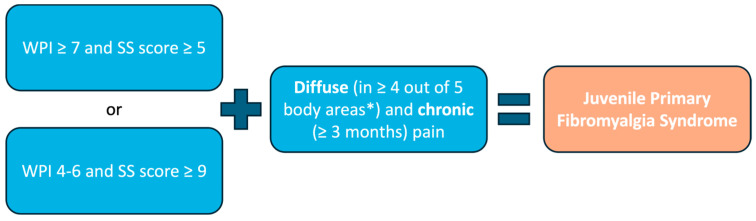
Diagnostic algorithm for Juvenile Primary Fibromyalgia Syndrome. * Region 1 = left upper region (left shoulder girdle, upper and lower arm); region 2 = right upper region (right shoulder girdle, upper and lower arm); region 3 = left lower region = left hip, upper and lower leg; region 4 = right lower region (right hip, upper and lower leg); region 5 = axial region (neck, upper and lower back).

**Figure 2 biomedicines-13-01168-f002:**
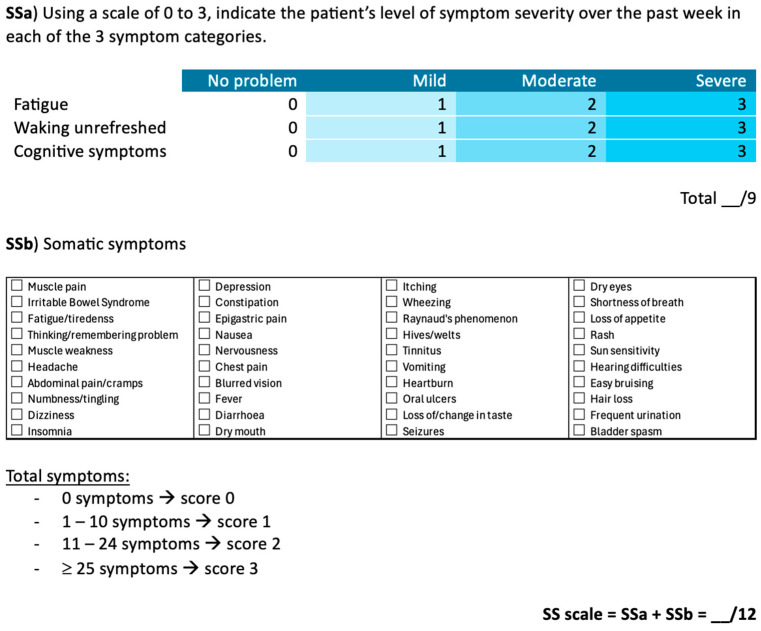
Symptom Severity (SS) Scale [72].

**Table 1 biomedicines-13-01168-t001:** Yunus and Masi proposed diagnostic criteria for Juvenile Primary Fibromyalgia Syndrome (JPFS) [1]. A diagnosis of JPFS is made when all major criteria are met, along with at least three minor criteria, or when there are four tender points and five minor criteria.

**Major Criteria**
Generalized musculoskeletal aching at ≥3 sites for ≥3 months
Absence of an underlying condition or cause Normal laboratory tests
≥5 typical tender points ^1^
**Minor Criteria**
Chronic anxiety or tension
Fatigue
Poor sleep
Chronic headaches
Irritable bowel syndrome
Subjective soft tissue swelling
Numbness
Pain modulated by physical activity
Pain modulated by changes in weather
Pain modulated by anxiety or stress

^1^ Yunus and Masi initially listed 31 tender points. However, the 1990 American College of Rheumatology criteria later suggested 18 tender points [71] (occiput, low cervical, trapezius, supraspinatus, second costochondral junction, lateral epicondyles, gluteal folds, posterior greater trochanter, and the medial knee fat pad).

**Table 2 biomedicines-13-01168-t002:** Widespread Pain Index [72,73]. Mark each area where the patient has experienced pain in the past week; each marked area contributes 1 point to the total score. The WPI ranges from 0 to 19.

Left Region	Right Region	Axial Region
Left upper region Jaw, left Shoulder girdle, left Upper arm, left Lower arm, left	Right upper region Jaw, right Shoulder girdle, right Upper arm, right Lower arm, right	Neck Upper back Lower back Chest Abdomen
Left lower region Hip (buttock, trochanter), left Upper leg, left Lower leg, left	Right lower region Hip (buttock, trochanter), right Upper leg, right Lower leg, right	

**Table 3 biomedicines-13-01168-t003:** Summary of current evidence on treatment approaches for Juvenile Primary Fibromyalgia Syndrome (JPFS).

Therapeutic Strategy	Study Type	Cohort (n)	Description	Evidence of Effectiveness
Aerobic exercise	RCT [92]	30	30-min cardio-dance and boxing, 3×/week for 12 weeks	Significant improvements in physical function, capacity, quality of life, fatigue
CBT	RCT [93]	114 (57/group)	8 weekly 45-min sessions (parents in 3/8 sessions) vs. education	Significant reduction in functional disability and depression
CBT + neuromuscular exercises (FIT Teens)	RCT [94,95]	40 (20/group)	16 small-group sessions (2×/week for 8 weeks) vs. CBT alone	Superior to CBT alone in reducing pain and improving strength/biomechanics
Pregabalin	RCT [96]	107 (54 pregabalin, 53 placebo)	15-week flexible-dose pregabalin (75–450 mg/day) vs. placebo	No significant improvement in mean pain score at endpoint
Duloxetine	RCT [97]	184 (91 duloxetine, 93 placebo)	13-week course (30–60 mg QID) vs. placebo	No significant change in pain severity, but greater overall treatment response
Milnacipram	Open-label trial [98]	116 (20 randomized post-response)	8-week open-label phase (≤100 mg/day), then 8-week milnacipram vs. placebo	Improvement in pain, quality of life, anxiety; early study termination
Fluoxetine	Open label trial [99]	10 (4 completed full protocol)	12-week flexible-dose (10–60 mg/day)	Reduction in pain and overall JPFS impact; very limited sample

Abbreviations: CBT, cognitive-behavioral treatment; FIT Teens, Fibromyalgia Integrative Training program for Teens; QID, four times a day; RCT, randomized controlled trial.

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
