# Peer review of "Juvenile Primary Fibromyalgia Syndrome: Advances in Etiopathogenesis, Clinical Assessment and Treatment: A Narrative Review"

_biomedicines, 2025, doi:10.3390/biomedicines13051168_

Round 1
Reviewer 1 Report
Comments and Suggestions for Authors
Dear authors
Title:
Could you add the type of study?
Abstract:
Nothing to state
Introduction:
Could you expand a bit more your introduction, giving more information about his condition, and then explain why your study is interesting?
Methods:
Which was your search strategy? Even if this is a narrative review, with the aim of increasre transparency, we need to include the search strategy, which database were consulted, which were your inclusion/exclusion criteria, etc.
Others:
Could you speak a bit more about the evidence of treatment strategies, including a table, etc?
Could you include a deeper critical analysis of the literature? Many subthemes are included, but the reader should know which is the start of the evidence for each of your arguments. You could also speak deeply about limitations in the management of these patients.
Conclusions:
Your conclusions should be shorter and be based on what is demonstrated with strong evidence. What is not demonstrated with strong evidence could be included as lines for future research.
Author Response
Title: Could you add the type of study?
We thank the reviewer for this helpful suggestion. We have now included the type of study in the title, as requested (see page 1 of the revised manuscript).
Abstract: Nothing to state
We appreciate the reviewer’s positive feedback and are pleased that the abstract was satisfactory.
Introduction: Could you expand a bit more your introduction, giving more information about his condition, and then explain why your study is interesting?
We thank the reviewer for this valuable comment. We have expanded the introduction to include additional information on the epidemiology of Juvenile Primary Fibromyalgia Syndrome (JPFS) (see pages 1 and 2). We have also clarified the rationale and significance of our review by refining the aims of the study (page 2).
Methods: Which was your search strategy? Even if this is a narrative review, with the aim of increasre transparency, we need to include the search strategy, which database were consulted, which were your inclusion/exclusion criteria, etc.
We thank the reviewer you for this observation. To enhance transparency, we have now included a detailed description of our search in the Methods section (page 2). We conducted a literature search in the PubMed database in February 2025 focusing on the past 20 years. The following search terms were used: “juvenile primary fibromyalgia syndrome” and combined searches of “fibromyalgia” AND “children”, “fibromyalgia” AND “child”, “fibromyalgia” AND “paediatrics”. We also examined the reference lists of relevant articles to identify additional sources.
Only peer-reviewed articles published in English between January 2005 and February 2025 were included. Exclusion criteria were: non-English publications, articles not available in full text, and publication types such as case reports, editorials, letters to the editor, and conference abstracts.
Others: Could you speak a bit more about the evidence of treatment strategies, including a table, etc? Could you include a deeper critical analysis of the literature? Many subthemes are included, but the reader should know which is the start of the evidence for each of your arguments. You could also speak deeply about limitations in the management of these patients.
We thank the reviewer for this insightful comment. In response, we have expanded our discussion of treatment strategies, providing further details on the current evidence base (see pages 10, 11 and 14). To enhance clarity and accessibility, we have also added Table 3, which summarises the available evidence on various therapeutic approaches evaluated in paediatric populations. Additionally, we have addressed key limitations in the management of JPFS, particularly the challenges in maintaining adherence to physical activity, often due to pain exacerbation during exercise and the associated fear of movement (see pages 14 and 15). We hope that these revisions contribute to a clearer understanding of the treatment landscape for JPFS.
Conclusions: Your conclusions should be shorter and be based on what is demonstrated with strong evidence. What is not demonstrated with strong evidence could be included as lines for future research.
We appreciate the reviewer’s suggestion. In response, we have shortened the conclusions by omitting the discussion on small fiber neuropathy, which currently lacks strong supporting evidence. We have focused the conclusions on findings supported by robust evidence, while highlighting areas in need of further investigation for future research.
Reviewer 2 Report
Comments and Suggestions for Authors
This is a well written manuscript. It starts with a gentle introduction to the Juvenile Primary Fibromyalgia Syndrome (JPFS). The introduction is concise and up to the point, but it would be good to discuss why JPFS is usually affects girls? Perhaps, boys are more stable emotionally?
I also like how the authors describe categories of pain - nociceptive, neuropathic, and nociplastic. Definitions are given and it is easy to follow. I believe that the authors described all possible factors of JPFS. Major and minor criteria for JPFS as well as widespread pain index have also been discussed.
Pharmacological and non-pharmacological treatment (e.g., vigorous aerobic exercise for at least 30 minutes, two to three times weekly, and psychological therapies) have also been discussed. Besides them, the authors reviewed supplementary and physical therapies (such as Transcutaneous Electrical Nerve Stimulation (TENS)). TENS should be a useful therapy for FM patients as it decreases morphine requirements, incidence of postoperative nausea and vomiting, dizziness, and pruritus as discussed in a recent meta-analysis (https://pubmed.ncbi.nlm.nih.gov/38256561/). The authors could also mention the effect of a pulsed electromagnetic field (PEMF) in managing fibromyalgia patients (https://ieeexplore.ieee.org/document/10836606?signout=success). Overall, the authors did a very good job and there is almost no space for improvement.
Author Response
This is a well written manuscript. It starts with a gentle introduction to the Juvenile Primary Fibromyalgia Syndrome (JPFS). The introduction is concise and up to the point, but it would be good to discuss why JPFS is usually affects girls? Perhaps, boys are more stable emotionally?
Thank you for your comment. We have expanded the discussion on sex differences in response to pain, which is a relevant aspect of JPFS. Current literature suggests that males and females exhibit different pain responses, with females generally showing greater pain sensitivity and an increased risk of clinical pain. However, studies have also shown that males may experience more severe pain, heightened pain-related distress, and a tendency toward catastrophic thinking about pain. Furthermore, higher levels of depression and anxiety have been documented more frequently among males in clinical settings. We believe that these factors may help explain the sex differences observed in the prevalence of JPFS. This discussion has been added to the manuscript on page 1.
I also like how the authors describe categories of pain - nociceptive, neuropathic, and nociplastic. Definitions are given and it is easy to follow. I believe that the authors described all possible factors of JPFS. Major and minor criteria for JPFS as well as widespread pain index have also been discussed.
Thank you very much for your kind comment. We are glad that you appreciated the description of different types of pain and the overview of etiopathogenesis and diagnosis. We aimed to provide a clear and accessible explanation of nociceptive, neuropathic, and nociplastic pain, and we are pleased that it was easy to follow. We also appreciate your recognition of the comprehensive discussion of the major and minor criteria for JPFS, as well as the widespread pain index.
Pharmacological and non-pharmacological treatment (e.g., vigorous aerobic exercise for at least 30 minutes, two to three times weekly, and psychological therapies) have also been discussed. Besides them, the authors reviewed supplementary and physical therapies (such as Transcutaneous Electrical Nerve Stimulation (TENS)). TENS should be a useful therapy for FM patients as it decreases morphine requirements, incidence of postoperative nausea and vomiting, dizziness, and pruritus as discussed in a recent meta-analysis (https://pubmed.ncbi.nlm.nih.gov/38256561/). The authors could also mention the effect of a pulsed electromagnetic field (PEMF) in managing fibromyalgia patients (https://ieeexplore.ieee.org/document/10836606?signout=success). Overall, the authors did a very good job and there is almost no space for improvement.
Thank you again for your positive feedback on the section related to treatment strategies. We greatly appreciate the articles you suggested on TENS and PEMF. After reviewing them, we found them very insightful and have now incorporated them into our manuscript (see page 14).